# α-Methylacyl-CoA Racemase from *Mycobacterium tuberculosis*—Detailed Kinetic and Structural Characterization of the Active Site

**DOI:** 10.3390/biom14030299

**Published:** 2024-03-02

**Authors:** Otsile O. Mojanaga, Timothy J. Woodman, Matthew D. Lloyd, K. Ravi Acharya

**Affiliations:** Department of Life Sciences, University of Bath, Claverton Down, Bath BA2 7AY, UK; oom21@bath.ac.uk (O.O.M.); tw226@bath.ac.uk (T.J.W.)

**Keywords:** α-Methylacyl-CoA racemase (AMACR, P504S), CoA-transferase, colorimetric assay, epimerase, fenoprofen, ibuprofen, lipid metabolism, *M. tuberculosis*, X-ray crystallography

## Abstract

α-Methylacyl-CoA racemase in *M. tuberculosis* (MCR) has an essential role in fatty acid metabolism and cholesterol utilization, contributing to the bacterium’s survival and persistence. Understanding the enzymatic activity and structural features of MCR provides insights into its physiological and pathological significance and potential as a therapeutic target. Here, we report high-resolution crystal structures for wild-type MCR in a new crystal form (at 1.65 Å resolution) and for three active-site mutants, H126A, D156A and E241A, at 2.45, 1.64 and 1.85 Å resolutions, respectively. Our analysis of the new wild-type structure revealed a similar dimeric arrangement of MCR molecules to that previously reported and details of the catalytic site. The determination of the structures of these H126A, D156A and E241A mutants, along with their detailed kinetic analysis, has now allowed for a rigorous assessment of their catalytic properties. No significant change outside the enzymatic active site was observed in the three mutants, establishing that the diminution of catalytic activity is mainly attributable to disruption of the catalytic apparatus involving key hydrogen bonding and water-mediated interactions. The wild-type structure, together with detailed mutational and biochemical data, provide a basis for understanding the catalytic properties of this enzyme, which is important for the design of future anti-tuberculosis drug molecules.

## 1. Introduction

α-Methylacyl-CoA racemase (AMACR), also known as P504S, is an enzyme that plays a vital role in branched-chain fatty acid metabolism [1,2]. It catalyses the racemization of α-methyl-branched fatty acyl-CoAs, converting the (2*S*)-epimer to a near racemic mixture of the (2*R*)- and (2*S*)-epimers.

In *Mycobacterium tuberculosis* (*M. tuberculosis*), the causative agent of tuberculosis (TB), α-methylacyl-CoA racemase is known as MCR (with 44% amino acid sequence identity with the human AMACR enzyme), and is of particular interest due to its involvement in cholesterol metabolism. Cholesterol is an essential component of the host cell membrane and represents a potential carbon and energy source for *M. tuberculosis* during infection. The enzyme enables the conversion of α-methyl-branched fatty acids derived from cholesterol metabolism into forms that can be utilized by *M. tuberculosis* for growth and persistence within the host, a key component of *M. tuberculosis* pathogenesis [3].

Understanding the structure–function relationships of MCR in *M. tuberculosis* has implications for the development of novel therapeutic strategies against TB. Targeting MCR presents an attractive approach to disrupting the bacterium’s ability to utilize cholesterol and impair its survival within the host. Inhibiting MCR activity could potentially starve *M. tuberculosis* of essential nutrients, rendering it more susceptible to host immune responses and antimicrobial treatments. Furthermore, the structure–function relationships of MCR in *M. tuberculosis* can guide the development of selective inhibitors that specifically target the bacterium while sparing the human counterpart. Exploiting these differences can help design antimicrobial agents that specifically inhibit MCR in *M. tuberculosis* without affecting human AMACR (a drug target for prostate cancer [1,2,4,5,6,7,8]), minimizing potential side-effects. Acyl-CoA inhibitors of MCR have been reported [9,10].

Previously, the crystal structure of MCR in the wild-type form (1.8 Å) and ligand-bound complexes (2–2.5 Å) have been reported, which have provided the general features of the enzyme molecule and some important structural insights on the active site and ligand interactions [11,12,13]. The crystal structure of the apo-form of another fatty acyl-CoA-racemase from *M. tuberculosis* has also been reported [14,15]. Here, we report the crystal structure of MCR in a new crystal form (the enzyme was expressed and purified using a modified protocol in *Escherichia coli* (*E. coli*) [16]) followed by the detailed structural and kinetic characterization of three key active-site mutants (H126A, D156A and E241A) with the aid of high-resolution crystal structures and a modified colorimetric assay, respectively. These results provide a comprehensive picture of the MCR active site and molecular details of critical active-site residues in MCR.

## 2. Materials and Methods

General laboratory reagents were purchased from Merck (Gillingham, Dorset, UK) or Fisher Scientific (Loughborough, Leicestershire, UK) and used without further purification. Molecular biology reagents were obtained from New England BioLabs (Hitchen, Hertfordshire, UK), Stratagene, Promega (Southampton, Hampshire, UK) or Novagen (Madison, WI, USA). The four pET3a plasmids for wild-type MCR and the three mutants H126A, D156A and E241A were synthesized through Genewiz, UK (Takeley, UK). Protein columns and resins were obtained from Cytiva Life Sciences (Little Chalfont, Buckinghamshire, UK). All reagents were procured through UK suppliers or agents. Colorimetric substrate was synthesised as previously described [17].

### 2.1. Protein Expression and Purification

*M. tuberculosis* recombinant MCR was expressed and purified as described previously [16]. Briefly, MCR encoding the pET3a plasmid was transformed into a competent *E. coli* BL21 (DE3) pLysS host Novagen (Madison, WI, USA), which was used for the expression of the target protein with 0.4 mM isopropyl-β-d-galactopyranoside (IPTG) induction. MCR enzyme was purified from the cell lysate using diethylaminoethyl- (DEAE-) anion-exchange chromatography coupled with a 0 to 0.6 M NaCl gradient. This was followed by RESOURCE-Q anion exchange chromatography in combination with a 0 to 0.6 M NaCl gradient. Sephacryl S-100 size-exclusion chromatography at pH 7.5 was used as the last purification step. A total of 46 mg of MCR was purified from 10 g of cell mass, and the enzyme was buffer-exchanged into 10 mM potassium phosphate, pH 8.8, and concentrated to 6 to 7 mg/mL for storage at −80 °C until further use. The same protocol was used for the expression and purification of three MCR active-site mutants. The purity of wild-type MCR and the mutants was checked by sodium dodecyl sulfate polyacrylamide gel electrophoresis (SDS-PAGE), liquid chromatography–mass spectrometry (LC/MS) analysis, analytical size-exclusion chromatography, trypsin digestion and dynamic light scattering prior to structural and kinetic characterization.

### 2.2. ^1^H NMR Analysis

Unlabelled 2*S*- or 2*R*-fenoprofenoyl-CoA substrate (100 µM final concentration) was incubated with MCR (54 ng) in 50 mM NaH_2_PO_4_-NaOH pH 7.2 supplemented with ca. 87% (*v*/*v*) ^2^H_2_O, in a total volume of 550 µL at 30 °C for 1 h [18]. This mixture, along with a negative control from 85 °C heat-inactivation of the enzyme, was incubated in a 55 °C water bath for 10 min. Once cooled, the 550 µL solutions were added to 10 mm glass ^1^H NMR tubes, and 500 MHz ^1^H NMR spectral data were collected at 24 °C using a Bruker Avance III NMR spectrometer (Bruker, Massachusetts, U.S.A.). Spectra were referenced to residual water and were processed using Bruker Topspin 4.0.7. following the described protocol [18].

### 2.3. Colorimetric Assay

A modified protocol from the previously reported assay based on the elimination of 2,4-dinitrophenolate (Appendix A) was used to measure the activity and kinetics of MCR [17]. Concentrated substrate stock solutions were vortex-mixed for 5 min before dilution, and working substrate stocks were vortex-mixed for 2 min before use. Assays were conducted in half-volume microtitre plates in 50 mM NaH_2_PO_4_-NaOH, 100 mM NaCl and 1 mM ethylenediaminetetraacetic acid (EDTA) at pH 7.4 in a total volume of 100 µL. For enzyme titration and kinetic experiments, 2× stock of the enzyme and substrate were mixed. For additive experiments, 4× stocks of the enzyme and additive were mixed, and the assay was initiated by adding 2× stock of the substrate. Reactions were monitored for 3 or 6 min at 30 °C and 354 nm using a BMG Labtech Omega plate reader. Rates were determined using ICEKAT [19] using absorbance readings over the required time course, taking into account delays between adding the substrate and initiating readings. Rates were converted to µmol.min^−1^.mg^−1^ using the 2,4-dinitrophenoxide extinction coefficient (15.3 mM^−1^.cm^−1^) and path length (0.588 cm) [17]. These rate data were analysed using SigmaPlot 14.5 (Systat). Kinetic data were analysed using direct (Michaelis–Menten), direct linear, Lineweaver–Burk, Hanes–Woolf, Eadie–Hofstee, and residual plots using the enzyme kinetics macros, and the *K*_m_ and *V*_max_ values determined. Values for *k*_cat_ and *k*_cat_/*K*_m_ were calculated assuming an MCR M_w_ of 39,121, 39,055, 39,077 and 39,063 Da for the wild-type and H126A, D156A, and E241A mutants, respectively (determined using mass spectrometry, Appendix A).

### 2.4. X-ray Crystallography

Protein crystallization was carried out using the sitting drop and hanging vapour diffusion method at 22 °C. An Art Robbins Phoenix crystal screening nano-dispenser was used to set up MCR (at 6–7 mg/mL concentration) wild-type and mutant with sitting drop crystallization trials on 96-3 crystallization intelli-plates (Molecular Dimensions, Sheffield, UK) with the following screens: BCS, Morpheus I, PACT, Proplex. Within each well, a 2:1 and 1:1 ratio of protein solution to reservoir was mixed and 50 µL of reservoir solution added to the well. Crystals were observed after 4 weeks.

For optimization, using 24-well plates, hanging drop vapour diffusion crystallization trials were set up, with either 6–7 mg/mL MCR (for both wild-type and mutant forms of the enzyme) and 1.52 M di-ammonium phosphate-phosphoric acid with 10 mM BaCl_2_, at pH 7.0, in a 1 to 1 ratio, or 0.03 M magnesium chloride hexahydrate, 0.03 M calcium chloride dihydrate, 20 % (*v*/*v*) ethylene glycol, 10 % (*w*/*v*) PEG 8000 and 0.1 M sodium 4-(2-hydroxyethyl)piperazine-1-ethanesulfonic acid/3-(*N*-morpholino)propanesulfonic acid (HEPES-MOPS) at pH 7.5 in a 2 to 1 ratio. The resultant 3 and 4 µL drops were incubated with 1 mL reservoir solution, and crystals usually appeared after 3–4 weeks.

MCR crystals (both wild-type and mutants) were flash-frozen and stored at 100 K in liquid nitrogen until data collection at the Diamond synchrotron beamlines I03 or I04 (Diamond Light Source, Didcot, UK). Crystals were kept at constant temperature (100 K) under the liquid nitrogen jet during data collection. Images were collected using an Eiger2 XE 16M detector (Dectris, Baden-Daettwil, Switzerland). For each crystal, 3600 X-ray diffraction images were collected at a 0.1° oscillation, 0.002 s exposure time and 14.00 MGy dose. The best MCR diffraction data sets were indexed and integrated using DIALS, and scaled using AIMLESS (CCP4i2 suite) [20,21,22]. The solvent content within the crystal was estimated using Matthew’s coefficient followed by molecular replacement in PHASER using the previously reported MCR crystal structure (PDB code: 1X74) [11]. The molecular graphics software COOT was used for rounds of manual model building, while REFMAC5 was used for constrained refinement and the addition of water molecules [23,24]. The refined model was validated using Molprobity and PDB validation [21,25]. All structural illustrations were prepared using CCP4mg [26].

## 3. Results and Discussion

### 3.1. Biophysical Characterization of the Purified Wild-Type MCR and Three Active-Site Mutants

Large quantities of wild-type MCR and three active-site mutants (H126A, D156A and E241A) were expressed and purified (Appendix A). Detailed characterization of these enzymes by mass spectrometry showed that each contained an additional Met residue at the N-terminus and a short extension of amino acids (with the sequence GSGC) at the C-terminus (Appendix A). The dynamic light scattering experiments showed these protein samples were monodispersed (Appendix A). Analytical size-exclusion chromatography revealed that each protein assumed a homodimer conformation in solution (Appendix A). Collectively, these biophysical results confirmed the suitability of the expressed proteins for protein crystallization, followed by a detailed X-ray crystallographic study.

### 3.2. Enzymatic Activity Assays

A ^1^H NMR assay was used to determine whether the purified wild-type MCR was able to catalyse the expected epimerization reaction. Both active and heat-inactivated MCR were incubated with fenoprofenoyl-CoA and ^2^H_2_O, and then, ^1^H NMR spectra were collected. The spectrum of the heat-inactivated negative control showed a doublet between δ = 1.3 and 1.4 ppm, which corresponds to the signal for the C_α_-Me (carbon-2) group of fenoprofenoyl-CoA (Figure 1B) with a C_α_-^1^H. In contrast, the ^1^H NMR spectrum for the active MCR had its doublet collapse into a single peak (a coalesced triplet with *J* = ~1 Hz) upon replacement of the C_α_-^1^H with C_α_-^2^H (Figure 1A) [27]. This ^2^H wash-in showed that the purified MCR was an active dimer. α-Proton exchange is an obligatory step which is required by the enzyme when catalysing the physiological epimerization reaction. This ^2^H wash-in is consistent with a ^1^H NMR report where active human AMACR 1A exchanged C_α_-^1^H in fenoprofenoyl-CoA for ^2^H [28].

Fenoprofenoyl-CoA, along with several other analogues which include ibuprofenoyl-CoA, is a well-characterized AMACR substrate and inhibitor (IC_50_ value of fenoprofenoyl-CoA: 400 nM [17,29,30]). It has been shown previously that MCR was able to catalyse ibuprofenoyl-CoA epimerization, demonstrating similar catalytic features in comparison with AMACR [5,17,31,32]. A similar ^1^H NMR spectrum (400 MHz) approach showing a change in methyl group coupling upon AMACR-catalysed α-proton exchange has been used with (2*S*)- and (2*R*)-methyldecanoyl-CoA, although the peak for C_α_-Me was at δ = 1.00 [29,33]. Through derivatization, the ratio of *R*- and *S*- products could be calculated, thereby turning this assay into a measure of epimerization activity.

A colorimetric assay based on the elimination of the yellow chromophore 2,4-dinitrophenolate (ε_354_ = 15,300 M^−1^cm^−1^) from the racemic substrate 3-(2,4-dinitrophenoxy)-2-methylpropanoyl-CoA (Appendix A) was developed to assay AMACR [17]. This assay was adapted and optimized to measure MCR catalytic activity. Reaction volumes of 100 µL were setup by incubating the substrate with active wild-type or one of the three mutants in microtitre plates, and the change in absorbance at 354 nm was monitored. An intense yellow colour appeared in an enzyme concentration-dependent manner within the positive control and test samples, but not in the heat-inactivated negative controls. This colour change was a visual cue that the MCR was able to eliminate the product.

To apply and optimize this assay for MCR, several concentrations of wild-type MCR were used; then, the effect of four additives on MCR wild-type activity was investigated. Plots of absorbance against time were made and show that wild-type MCR was significantly more active than the AMACR control (by ca. 2010-fold) and the three MCR mutants (by ca. 12.5-, 2.5- and 2.5-fold for the H126A, D156A and E241A mutants, respectively) (Figure 2). The time-corrected absorbance data along with the extinction coefficient of the yellow product were used to calculate the specific activity and reaction rate of wild-type MCR and its three mutants. The rate-against-amount of enzyme plots (Figure 2) show the expected enzyme-dependent increase in rate, while the specific activities calculated for wild-type, H126A, D156A and E241A were 46.4, 3.2, 18.5 and 12.2 µmol.min^−1^.mg^−1^ (Figure 2E). Based on these specific activities, it was observed that the wild-type MCR enzyme was more active than all the H126A, D156A and E241A mutants. In contrast, previous studies [11] showed that these three active-site mutants had <0.2% activity of the wild-type enzyme using a radiochemical assay. The reasons for these different observations of the active-site mutants using different assays are unclear, but it may be due to the presence of a substantial tritium kinetic isotope effect when measuring activity using the reported ^3^H-release assay [11].

The colorimetric assay can be used on MCR to measure its activity in a concentration-dependent manner, but it is important to investigate whether this could be disrupted by additives. These additives can range from polar solvents used to enhance inhibitor solubility to detergents used to prevent the formation of aggregates [34]. As such, MCR is likely to encounter considerable concentrations of these additives when screening inhibitors. Using optimized conditions of 0.0122 µg wild-type MCR, 40 µM substrate, 50 mM NaH_2_PO_4_-NaOH, 100 mM NaCl and 1 mM EDTA, pH 7.4 and a 100 µL volume, reactions containing dimethylsulfoxide (DMSO), Triton X-100 or dithiothreitol (DTT) were set up and their absorbance at 345 nm measured over 5 min. These absorbance data (Figure 3A) showed an additive-dependent change in activity with increasing DMSO concentrations reducing the absorbance. The presence of DMSO produced a modest reduction in activity, while 0.01% *v*/*v* Triton X-100 and 1 mM DTT appeared to confer a slight increase in activity (Figure 3B). The impact of these additives on MCR activity was minimal. This robustness of MCR towards detergents is in line with AMACR, which showed no reduction in activity when 2-mercaptoethanol, dithiothreitol and bovine serum albumin (BSA) were present and a modest reduction in the presence of 0.1% (*v*/*v*) Triton X-100 [17].

### 3.3. Kinetic Assay

Following the investigation of the specific activity and tolerance of MCR to additives, the kinetic parameters for wild-type MCR and its mutants were analysed. Under optimized conditions, colorimetric substrates at six concentrations [150, 100, 66.7, 44.4, 29, 19.8, 13.2, 8.7 µM] were incubated with wild-type, H126A, D156A or E241A MCR. The rates for these data were calculated and fitted using a direct (Michaelis–Menten) plot (Appendix A) to determine the kinetic parameters. Based on the residual plots, there were no unexpected systematic trends in the data. Wild-type MCR was the most efficient enzyme with a *k*_cat_/*K*_m_ of 1.07 × 10^6^ M^−1^.s^−1^, while the H126A mutant was the least efficient at 0.11 × 10^6^ M^−1^.s^−1^ (Table 1). This near 10-fold decrease in catalytic efficiency for the H126A mutant relative to wild-type MCR contrasts with the near abolition of activity previously observed [11], but shows that although all three mutants had diminished activity, the H126A mutation was most detrimental. Rate data were also analysed using a direct linear plot, and the apparent *k*_cat_ and *K*_m_ values are reported (Appendix A). When comparing the determined MCR kinetic parameters with those reported for AMACR using the colorimetric assay [17], MCR was up to 702 × more efficient, as judged by *k*_cat_/*K*_m_ (Table 1).

In conclusion, ^1^H NMR using wild-type MCR demonstrated that the protein, when purified, was catalytically active and able to perform the physiological epimerization reaction. Using optimized conditions, the kinetic parameters of wild-type MCR and the three mutants were determined using a colorimetric assay. The MCR mutants had lower catalytic efficiency (judged by their *k*_cat_/*K*_m_) relative to wild-type MCR. Wild-type MCR had a catalytic efficiency of 1.07 × 10^6^ M^−1^.s^−1^, which represented a 702× increase in catalytic efficiency compared to human AMACR 1A [17]. The results show that the colorimetric assay is applicable to MCR. Enzymatic activity showed dose dependency, and all three mutants had diminished activity compared to wild-type MCR, which highlighted the importance of these three active-site residues for catalytic efficiency. Wild-type MCR maintained activity when incubated with DMSO, Triton-X100 and DTT.

### 3.4. Crystal Structure of Wild-Type MCR in a New Form

Crystals of wild-type MCR formed under a new condition [0.03 M magnesium chloride hexahydrate, 0.03 M calcium chloride dihydrate, 20 % (*v*/*v*) ethylene glycol, 10 % (*w*/*v*) polyethylene glycol (PEG 8000) and 0.1 M sodium HEPES-MOPS at pH 7.5] after 3 weeks of incubation at 22 °C, and these crystals diffracted to a 1.65 Å resolution and belonged to monoclinic space group C2 with a dimer in the asymmetric unit (Table 2). Several attempts were made to grow crystals using the previously reported condition (1.52 M di-ammonium phosphate-phosphoric acid and 10 mM barium chloride at pH 7.0) [11], but this did not yield any crystals, as the addition of 10 mM BaCl_2_ led to heavy precipitation. The new condition reported here was identified using a Morpheus screen (Molecular Dimensions, Sheffield, UK). The unit cell dimensions for this new crystal form, where a = 181.15 Å, b = 79.16 Å, c = 59.24 Å and β = 91.15˚ and the diffraction resolution cut-off is 1.65 Å, which were selected based on the CC_1/2_ value and the number of unique reflections present in the structure [20]. The previously reported crystal structure of wild-type MCR (PDB code: 1X74; [11]) had cell dimensions of a = 180.1 Å, b = 79.8 Å, c = 117.6 Å and β = 92˚ in monoclinic space group C2. However, due to doubling the of c-dimension (compared with the newly reported crystal form here), there were two MCR homodimers in the asymmetric unit. This could be attributed to the different crystallization conditions used. A comparison of the dimer arrangement in the two structures showed an identical arrangement of monomers (subunits).

The electron density maps for both monomers within the MCR homodimer were of high quality, except for residues 41 to 44, which form part of a flexible loop region. The overall structure of wild-type MCR (in the new crystal form) and the crystallographic statistics are shown in Figure 4A and Table 2, respectively. Each monomer (subunit) within the wild-type MCR dimeric structure has a large and a small domain; the large domain consists of residues M1 to A188 and R331 to G360, which come together to form an open α/β sheet structure with a Rossmann-like fold. This domain is composed of alternating β-strands and α-helices where the β-strands are hydrogen-bonded to each other, forming an extended β-sheet, and the α-helices surround both faces of the sheet to produce a three-layered sandwich (Figure 4A).

The small domain has a distinct three-stranded antiparallel β-sheet core structure made up of five α-helices and five β-strands from residues V189 to P300. A surface representation of the MCR structure (Figure 4A) shows the extended binding groove, which is made up of residues from the large domain of one subunit and the smaller domain of another subunit. The previously reported MCR structure [11] and the present structure, when superimposed (Figure 4B), had RMS deviations of 0.58 Å and 0.33 Å over 691 C_α_ atoms for each dimer in the asymmetric unit, indicating that these two crystal structures are nearly identical.

The MCR dimer has two active sites. Within each active site, the large domain contributes the catalytic residues H126 (part of α-helix 3) and D156 (part of α-helix 5), while the small domain contributes a third catalytic residue, E241 (part of α-helix 9). Within each active site, the side-chain of D156 sits parallel to H126 (from one monomer), which is hydrogen-bonded to residue E241 from the second monomer (Table 3, Figure 5A). Sandwiched between the D156 and H126 residues, a piece of electron density corresponding to part of a polyethylene glycol molecule (from the crystallization medium) was observed. Interestingly, a similar piece of electron density was observed in the previously reported MCR structure (PDB code: 1X74; [11]), which was attributed to a bound glycerol molecule. A network of bound water molecules was also observed within the active site due to the absence of a bound acyl-CoA ester (Figure 5A). A N-terminal methionine residue was modelled into the structure, but there was no density for the C terminal end, suggesting that the four additional residues were very mobile.

### 3.5. Crystal Structures and Hydrogen Bond Interactions of Three MCR Mutants at the Active Site

Crystals of the three MCR mutants H126A, D156A and E241A were obtained using identical conditions to those used with the wild-type enzyme and diffracted to 2.45, 1.64 and 1.85 Å resolutions, respectively. These three crystal structures had a similar space group, unit cell dimensions and number of molecules in the asymmetric unit. All three mutant structures had all the same MCR residues (except the mutated residue), including the extra N and C terminal residues, as the wild-type enzyme. The mutation of side-chains to alanine was unambiguously confirmed using the unrefined structures obtained from molecular replacement. The overall structures for these single mutants were very similar to those of the wild-type structure, with small root-mean-squared deviation (RMSD) differences of 0.38, 0.37 and 0.38 Å for 693 C_α_ atoms of the H126A, D156A and E241A mutant structures being observed, respectively.

Our detailed investigation of the hydrogen bond interactions of active-site residues (Figure 5B–D, Table 3) within one of the active sites of wild-type MCR showed 10 hydrogen bond interactions through water-mediated interactions, part of a polyethylene glycol molecule, and interactions involving other neighbouring residues. These hydrogen bond interactions were also observed within the second active site, which also had 10 interactions. The three mutants collectively had fewer hydrogen bond interactions; within the H126A, D156A and E241A mutants, 3, 5 and 5 interactions were observed, respectively. Among these active-site mutants, H126A had the fewest overall interactions, while D156A and E241A had a higher yet similar number of interactions. This observed difference in hydrogen bonding interactions aligns with the relative difference in enzymatic activities reported based on kinetic data, wherein a great number of hydrogen bonding interactions corresponds with high activity, although this relationship is non-linear.

Within the active sites of the three MCR mutants, the mutations of one of the three active-site residues to alanine resulted in subtle structural changes due to a loss of hydrogen bond interactions coupled with minor local structural rearrangements to stabilize affected residues. The active site of the H126A mutant underwent the most profound structural change due to the loss of the imidazole ring, although D156 and its hydrogen bond interactions remained unaffected (Figure 5B). The E241 side-chain which is hydrogen-bonded to H126 within the wild-type, is rotated by around 90° and projects away from the active site due to loss of its binding partner (H126). Interestingly, the normalized temperature factor (B-value, 45.79 Å^2^) of this E241 side-chain is lower than expected due to the peripheral Q243 (39.70 Å^2^) residue hydrogen bonding to it. As the E241 residue is stabilized by Q243, it moves away from D156, and hence, it cannot substitute the central structural role of the imidazole ring of H126. Consequently, five H126-mediated hydrogen bond interactions are lost. The observed displacement of E241 also suggests that it is unlikely to substitute the catalytic function of H126, which implies that the residual enzymatic activity observed in the H126A mutant is facilitated solely by the D156 side-chain.

The active site of the D156A mutant (Figure 5C) has the fewest structural changes, as the H126-and-D241 dyad is retained along with the associated hydrogen bonding interactions. Within the wild-type, the D156 side-chain is hydrogen-bonded to a PEG molecule and a water molecule (which form part of a network which terminates with Y130) and a peripheral N152 residue. Consequently, as this D156 side-chain is mutated to Ala, these interactions are lost. Interestingly, the orientation peripheral N152 residue is unaffected and can form a hydrogen bond with part of the PEG molecule compared to the wild-type and the two other mutants.

The active site of the E241A mutant (Figure 5D) stands out, as the relative positions of the active-site residues D156 and H126 are identical to the wild-type. These two active-site residues remain parallel to each other, and part of the PEG molecule is nestled in between them; consequently, most of the associated hydrogen bond interactions are retained. Within this mutant, the H126 residue is hydrogen-bonded to the peptide nitrogen atom of A241 through a water molecule, and due to this interaction, the movement of H126 gets restricted and it does not pivot away like E241 within the H126A mutant. This important structural water molecule within the E241A mutant active site resides where the functional group of the E241 side-chain was. However, the introduction of this structural water molecule does not replace all the hydrogen bonds lost in the active site.

## 4. Conclusions

This report describes a detailed kinetic analysis of wild-type enzyme and active-site mutants using a colorimetric assay. This allows for the proper quantification of enzyme kinetic behaviour using *k*_cat_/*K*_m_ values, and for comparisons of the data to be made. Moreover, the colorimetric assay is not subject to a kinetic isotope effect, which would complicate the analysis, unlike the data from the previous report [11]. The detailed active-site structural features outlined for the wild-type and three mutants show how subtle structural changes affect the enzymatic activity of MCR in *M. tuberculosis* and provide insights into some key components within the catalytic site and their functional significance. One such feature is the central role of the imidazole ring of the H126 side-chain. The findings reported will form a good basis for a detailed study of MCR inhibitors for the development of potential anti-TB drugs and could allow MCR to be used as a model for AMACR 1A in drug development. AMACR 1A is a prostate cancer drug target [1,2,4,5,6,7,8] and is an MCR homologue with 43% sequence identity [11], but it has no reported crystal structures and there are challenges associated with it production. Reported AMACR 1A inhibitors are rationally designed acyl-CoA esters which have some challenges limiting their application.

## Figures and Tables

**Figure 1 biomolecules-14-00299-f001:**
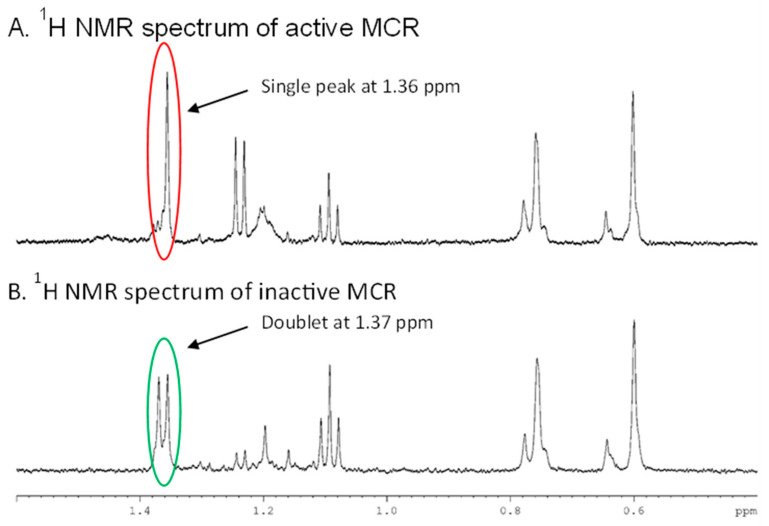
^1^H NMR spectra of wild-type MCR reactions. The ^1^H NMR spectra of incubated solutions containing 54 ng of wild-type MCR (active or heat-inactivated), 100 µM fenoprofenoyl-CoA and 50 mM NaH_2_PO_4_-NaOH pH 7.4 buffer with ca. 87% (*v*/*v*) ^2^H_2_O were collected. (**A**) Active MCR spectra. (**B**) Heat-inactivated MCR spectra. The peak at δ = 1.3–1.4 ppm corresponds to C_α_-Me. A doublet shows C_α_-H while a single peak shows C_α_-^2^H.

**Figure 2 biomolecules-14-00299-f002:**
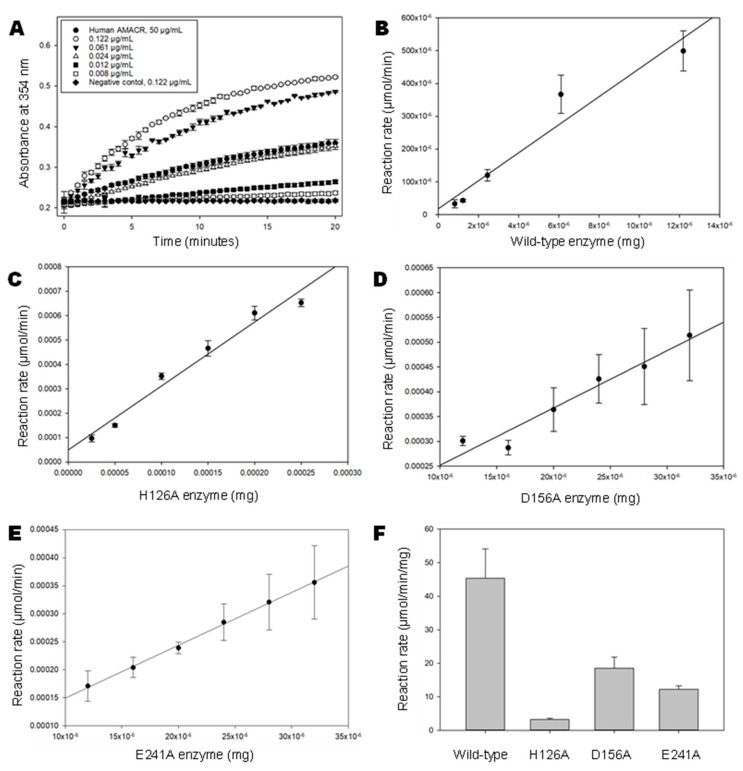
Colorimetric assay for wild-type MCR and its 3 active-site mutants. A total of 6 serial dilutions for MCR and its 3 mutants were separately incubated with colorimetric substrate in 100 µL volumes containing 40 µM colorimetric substrate and A_354_ were determined for 3 repeats over 10 min. ICEKAT was used to determine the activity of each enzyme [19]. (**A**) Formation of product by wild-type MCR over time. Human AMACR 1A [17] is shown for comparison. Data are presented as means ± SD; (**B**–**E**) Dependence of reaction rate on amount of enzyme for wild-type enzyme and H126A, D156A and E241A mutants, respectively. Data are presented as means ± SD; (**F**) Specific activity for wild-type (n = 4) and mutant MCR (n = 5). Data are presented as means ± SD.

**Figure 3 biomolecules-14-00299-f003:**
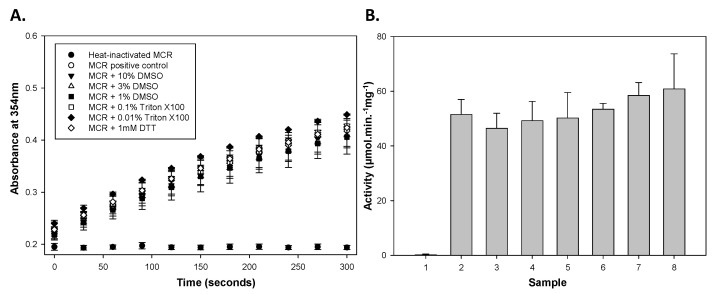
Graphs of absorbance against time and specific activity for wild-type MCR in the presence and absence of additives. We incubated 4× MCR stock with either 20, 12 or 4% (*v*/*v*) DMSO, 0.4 or 0.04 % (*v*/*v*) Triton X-100 or 4 mM DTT to give 2× enzyme and additive stock. 2× stock solution of substrate was added to this mixture, giving a final substrate concentration of 40 µM. A_354_ data over time was monitored and used to determine the specific activity (µmol.min^−1^.mg^−1^). (**A**) Absorbance against time. (**B**) Enzyme (0.2 µg/mL)-specific activity in the presence of additives, measured for 3 replicants (means ± SD). Samples: 1. Heat-inactivated MCR; 2. active MCR-positive control; 3. with 10% (*v*/*v*) DMSO; 4. with 3% (*v*/*v*) DMSO; 5. with 1% (*v*/*v*) DMSO; 6. with 0.1% Triton X-100; 7. with 0.01% Triton X-100; 8. with 1 mM DTT.

**Figure 4 biomolecules-14-00299-f004:**
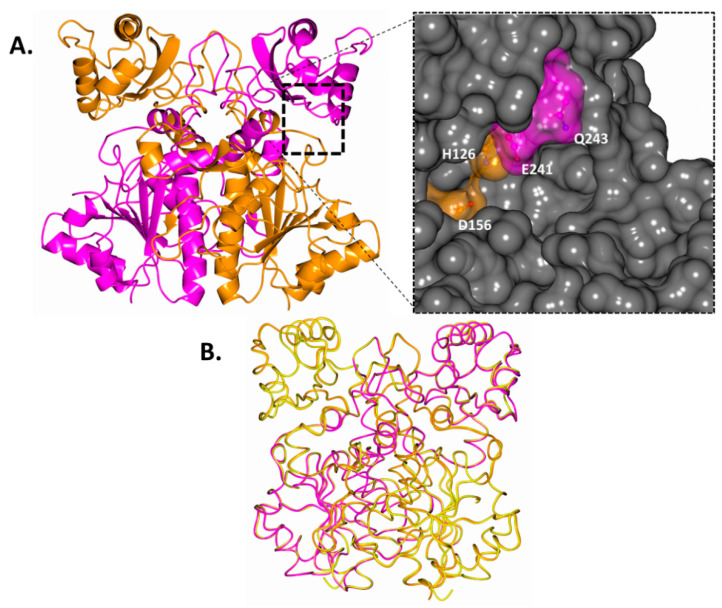
Structures of wild-type MCR. (**A**) Ribbon structure of the dimeric MCR. Subunit one: dark orange, subunit two: magenta. The 2 subunits are intertwined and form 2 active sites at opposite ends of the structure. The active site is made up of the large domain of one subunit and the small domain of the second-subunit. The inset presents the surface structure of the active site of wild-type MCR. For clarity, one active-site side-chain each from subunit 1 (dark orange) and subunit 2 (magenta) is shown. (**B**) Worm structure of the C_α_ backbone of wild-type MCR (subunit one: dark orange, subunit two: magenta) secondary structure matching (SSM) superimposed with subunits 1 and 2 (yellow) of the previously reported MCR structure (PDB code: 1X74 [11]). Images were prepared using CCP4mg.

**Figure 5 biomolecules-14-00299-f005:**
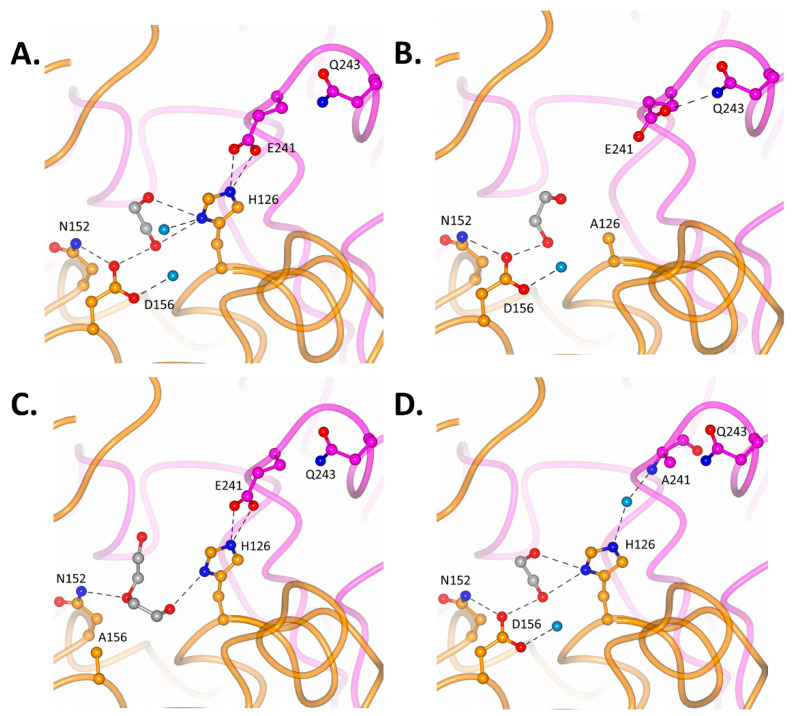
Detailed active-site structures of wild-type MCR and 3 of its mutants. Within these active sites, the MCR mutants show subtle structural changes. (**A**) The wild-type MCR catalytically active site where in the 3 catalytic residues H126, D156 and E241 are labelled, along with other important peripheral residues (subunit 1: dark orange and subunit 2: magenta). H126 and D156 residues are parallel to each other, while E241 is hydrogen-bonded to H126. Water molecules (light blue) and part of a bound polyethylene glycol molecule (dark grey) are shown. (**B**) Active site of H126A mutant shows a near 90° shift in E241 residue and forms a hydrogen bond with the peripheral Q243 side-chain. (**C**) The catalytically active site of the D156A mutant has more of the PEG molecule visible. (**D**) In the catalytically active site of the E241A mutant, the position and orientation of H126 resembles those of the wild-type MCR structure, and a water molecule is hydrogen-bonded to this H126 side-chain and the nitrogen from the backbone of A241.

**Table 1 biomolecules-14-00299-t001:** Kinetic parameters of wild-type MCR and its 3 mutants derived from fitting colorimetric assay data to the Michaelis–Menten equation using the direct plot. Data are presented as means ± SEM. Data for wild-type AMACR 1A were previously reported [17] and were derived using the direct linear plot.

Enzyme	*K*_m_ (µM)	Vmax (µmol.min^−1^.mg^−1^)	*k*_cat_ (s^−1^)	*k*_cat_/*K*_m_ (M^−1^.s^−1^)
MCR	96 ± 14	157 ± 13	102 ± 8.3	1.07 × 10^6^
MCR H126A	63.0 ± 5.5	11 ± 0.4	7.2 ± 0.3	0.11 × 10^6^
MCR D156A	70.0 ± 9.7	48 ± 3	31 ± 2.1	0.44 × 10^6^
MCR E241A	24.0 ± 3.1	23 ± 1	15 ± 0.7	0.63 × 10^6^
AMACR 1A [17]	58.0	0.112	0.088	1517

**Table 2 biomolecules-14-00299-t002:** X-ray crystallographic data collection and refinement statistics. Data for the outer shell are shown in parentheses.

	Wild-Type MCR	MCR H126A	MCR D156A	MCR E241A
Beamline	I04	I03	I04	I03
Wavelength used (Å)	0.9537	0.9762	0.9537	0.9762
Crystallographic statistics				
Space group	C2	C2	C2	C2
Unit-cell dimensions				
a, b, c (Å)	181.05, 79.09, 59.19	181.11, 79.15, 59.10	180.47, 78.75, 58.94	180.62, 78.99, 58.923
α, β, ϒ (°)	90.00, 92.01, 90.00	90.00, 91.66, 90.00	90.00, 92.151, 90.00	90.00, 92.19, 90.00
Resolution-range (Å)	90.47–1.65 (1.68–1.65)	90.52–2.45 (2.55–2.45)	90.17–1.64 (1.67–1.64)	90.24–1.85 (1.89–1.85)
R_merge_ (%)	0.098 (2.622)	0.422 (3.294)	0.100 (2.265)	0.139 (1.987)
R_pim_ (%)	0.040 (1.065)	0.159 (1.442)	0.041 (0.908)	0.057 (0.813)
CC_1/2_ (%)	0.999 (0.320)	0.982 (0.278)	0.998 (0.372)	0.998 (0.500)
Mean < I/σ(I) >	8.9 (0.7)	4.9 (0.8)	9.9 (0.9)	8.2 (0.9)
Completeness (%)	100 (100)	100 (100)	99.0 (98.2)	100 (100)
No. of observed reflections	679637 (33777)	214015 (20978)	688384 (34415)	478293 (30079)
No. of unique reflections	100251 (4908)	30851 (3454)	99692 (4868)	70688 (4349)
Multiplicity	6.8 (6.9)	6.9 (6.1)	6.9 (7.1)	6.8 (6.9)
Refinement statistics				
R_work_/R_free_	0.19/0.23	0.19/0.26	0.18/0.21	0.19/0.24
RMSD in bond lengths (Å)	0.011	0.016	0.016	0.016
RMSD in bond angles (°)	1.77	2.58	2.14	2.38
Ramachandran plot statistics (%)				
Favoured	96.29	94.00	96.86	94.29
Allowed	2.79	5.71	3.14	5.28
Outliers	0.92	0.29	0.0	0.43
Average B-Factors (Å^2^)				
Protein	33.31	48.5	29.84	33.59
Water	37.16	33.02	34.48	37.34
No. Atoms				
Protein	5387	5373	5380	5371
Water	421	61	429	413

**Table 3 biomolecules-14-00299-t003:** Hydrogen bond interactions of the residues in the active site of wild-type MCR and its three active-site mutants. The two subunits make up the MCR homodimer.

Subunit 1					
Donor	Acceptor	MCR Wild-Type	MCR H126A	MCR D156A	MCR E241A
His126 N^δ1^	PEG	3.18	-	2.96	3.08
	PEG	3.16	-	-	3.09
	Water	3.10	-	-	-
His126 N^ε2^	Glu241 O^ε1^/B	2.70	-	2.65	-
	Glu241 O^ε2^/B	3.24	-	3.23	-
	**Water**	-	-	-	**2.84**
Asp156 O^δ1^	Water	2.64	2.70	-	2.55
Asp156 O^δ2^	Asn152 N^δ2^	2.82	2.99	-	2.83
	PEG	2.78	2.51	-	2.77
**Ala241 N**	**Water**	-	-	-	**3.12**
Glu241 O^ε1^	His126 N^ε2^/B	2.65	-	2.61	-
Glu241 O^ε2^	His126 N^ε2^/B	3.27	-	3.25	-
	Water	3.04	-	3.17	-
**Subunit 2**					
**Donor**	**Acceptor**	**MCR** **Wild-Type**	**MCR H126A**	**MCR D156A**	**MCR E241A**
His126 N^δ1^	PEG	3.19	-	3.25	3.00
	PEG	-	-	-	3.30
	Water	3.14	-	-	-
His126 N^ε2^	Glu241 O^ε1^/A	2.65	-	2.61	-
	Glu241 O^ε2^/A	3.27	-	3.25	-
	**Water**	-	-	-	**2.80**
Asp156 O^δ1^	Water	2.72	-	-	2.57
Asp156 O^δ2^	Asn152 N^δ2^	2.87	2.99	-	2.98
	PEG	2.67	2.69	-	2.65
	Water	-	2.85	-	2.67
**Ala241 N**	**Water**				**3.00**
Glu241 O^ε1^	His126 N^ε2^/A	2.70	-	2.65	-
Glu241 O^ε2^	His126 N^ε2^/A	3.24	-	3.23	-
	Water	-		-	-
	Gln243 N^ε2^/B	-	3.07	-	-

Note—Hydrogen bond distances within the columns are in Å. For a hydrogen bond to be listed, the distance between the donor (D) and acceptor (A) must be shorter than 3.3 Å and the D-H-A angle greater than 120°. Donor and acceptor atoms are shown in bold due to the observation of a water molecule within the MCR E241A mutant active site.

## Data Availability

The atomic coordinates and structural factors for wild-type MCR and the three active-site mutant structures have been deposited with codes 8RMW, 8RP3, 8RP4 and 8RP5, respectively, in the RCSB Protein Data Bank, www.pdb.org (accessed on 21 January 2024).

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
