# Peer review of "α-Methylacyl-CoA Racemase from Mycobacterium tuberculosis—Detailed Kinetic and Structural Characterization of the Active Site"

_biomolecules, 2024, doi:10.3390/biom14030299_

Round 1

Reviewer 1 Report

Comments and Suggestions for Authors

The paper is interesting and quite well written however difficult
to follow due to many technical aspects.
This is not objection just statement.
Before the manuscript will be accepted a few aspects must be clarified.
1) I have found the paper entitled
Alpha-methylacyl-CoA racemase from Mycobacterium tuberculosis. Mutational and structural characterization of the active site and the fold written by Savolainen et al. published in 2005  https://pubmed.ncbi.nlm.nih.gov/15632186/
Its title and content is very similar. Moreover there is another not referenced paper written by Rhee et al, which discuss X-ray structure. https://www.ncbi.nlm.nih.gov/pmc/articles/PMC1978124/

2) Information about the origin of the sample (manufacturer site) from which chemical and biological reagents were purchased must be provided. 

3) The scale in Figure 1 is not given, so it is difficult to assess whether singlet and doublet can be compared. The term "doublets collapse" does not seem accurate.

4) The quality of the Figure 4 is poor, it is difficult to differentiate light and dark orange. Please change it.

5) The description of the molecules i.e. Molecule 1 and Molecule 2 is not adequate. Which one is which?

6) There is no information concerning the software used for analysis/visualization. 

7) The conclusions part must be rewritten , there is no information what was exactly done and how novel it is (please see objections in 1)).

To sum up:

The paper is not properly grounded in the literature. Reading manuscript  is difficult to assess what is new in this work, I have the impression that the authors avoid citing these papers despite they are very easy to find.

Comments on the Quality of English Language

Acceptable.

Reviewer 2 Report

Comments and Suggestions for Authors

The paper titled "α-Methylacyl-CoA racemase from Mycobacterium tuberculosis - a Detailed Kinetic and Structural Characterization of the Active Site" by Mojanaga and colleagues opens new horizons for α-Methylacyl-CoA racemase based on the beautiful structure of MCR. The authors' arguments are supported by extensive experimental evidence, including structural insights, and their data interpretation is technically very sound. This paper is suitable for publication in "Biomolecules," but the reviewer requests some minor revisions:

Minor issues:

1. The Introduction section, particularly the first paragraph on Page 1, is overly long. It is suggested to divide the discussion into sections focusing on AMACR and fatty acids.

2. The Methods section is generally well-described. However, more details are needed for section 2.4 X-ray Crystallography. Clarifications on experimental procedures, methods, and usage are necessary.

3. The data in the Supplementary Materials are of high quality and could be moved to the main manuscript as regular data. Consider relocating them for better visibility and accessibility.

Round 2

Reviewer 1 Report

Comments and Suggestions for Authors

I still think that a scale should be given on the Y axis in the NMR spectrum, just for the sake of the singlet/doublet comparison. (A doublet may appear as a result of quantum transitions for spin = 1, but also as a result of proton jump or sample contamination. Deuterated sample are often not stable.)
Replacing "molecule" term with "subunit" does not solve the problem of the lack of a clear description. It is not a problem of term but description.
I advise you to correct this before publishing. 
